# Impact of Genomic Mutation on Melanoma Immune Microenvironment and IFN-1 Pathway-Driven Therapeutic Responses

**DOI:** 10.3390/cancers16142568

**Published:** 2024-07-17

**Authors:** Fátima María Mentucci, Elisa Ayelén Romero Nuñez, Agustina Ercole, Valentina Silvetti, Jessica Dal Col, María Julia Lamberti

**Affiliations:** 1Departamento de Biología Molecular, INBIAS, Universidad Nacional de Río Cuarto, Río Cuarto X5800BIA, Argentina; fmentucci@exa.unrc.edu.ar (F.M.M.); valentinasilvetti@gmail.com (V.S.); 2Department of Medicine, Surgery and Dentistry, Scuola Medica Salernitana, University of Salerno, 84081 Baronissi, Italy; jdalcol@unisa.it

**Keywords:** BRAFV600E, IFN-1 pathway, cGAS-STING, immunogenic cell death

## Abstract

**Simple Summary:**

This study investigates the impact of the BRAFV600E mutation, present in half of melanoma cases, on immune microenvironment and therapeutic response. Analyzing the Cancer Genome Atlas data, we uncover that the mutation correlates with reduced tumor mutation burden, indicating a lower generation of immune-stimulating neopeptides. Examination of immune subtypes reveal heightened immunosuppression in BRAFV600E-mutated tumors. Using melanoma cell lines harboring different genomic profiles, we observed an enhanced response to immunogenic cell death (ICD)—a crucial process eliciting anti-tumor immune responses—mediated by photodynamic therapy (PDT). Transcriptomic analysis highlights upregulation of IFNAR1, IFNAR2, and CXCL10 genes in BRAFV600E-mutated cells, suggesting their role in ICD. Our results describe the intricate relationship between the BRAFV600E mutation and immune responses, hinting at a potential link between this mutation and responsiveness to ICD-inducing therapies, particularly PDT, putatively linked to increased IFN-1 pathway activation.

**Abstract:**

The BRAFV600E mutation, found in approximately 50% of melanoma cases, plays a crucial role in the activation of the MAPK/ERK signaling pathway, which promotes tumor cell proliferation. This study aimed to evaluate its impact on the melanoma immune microenvironment and therapeutic responses, particularly focusing on immunogenic cell death (ICD), a pivotal cytotoxic process triggering anti-tumor immune responses. Through comprehensive in silico analysis of the Cancer Genome Atlas data, we explored the association between the BRAFV600E mutation, immune subtype dynamics, and tumor mutation burden (TMB). Our findings revealed that the mutation correlated with a lower TMB, indicating a reduced generation of immunogenic neoantigens. Investigation into immune subtypes reveals an exacerbation of immunosuppression mechanisms in BRAFV600E-mutated tumors. To assess the response to ICD inducers, including doxorubicin and Me-ALA-based photodynamic therapy (PDT), compared to the non-ICD inducer cisplatin, we used distinct melanoma cell lines with wild-type BRAF (SK-MEL-2) and BRAFV600E mutation (SK-MEL-28, A375). We demonstrated a differential response to PDT between the WT and BRAFV600E cell lines. Further transcriptomic analysis revealed upregulation of IFNAR1, IFNAR2, and CXCL10 genes associated with the BRAFV600E mutation, suggesting their involvement in ICD. Using a gene reporter assay, we showed that PDT robustly activated the IFN-1 pathway through cGAS-STING signaling. Collectively, our results underscore the complex interplay between the BRAFV600E mutation and immune responses, suggesting a putative correlation between tumors carrying the mutation and their responsiveness to therapies inducing the IFN-1 pathway, such as the ICD inducer PDT, possibly mediated by the elevated expression of IFNAR1/2 receptors

## 1. Introduction

Melanoma, a malignant neoplasm originating from melanocytes, represents a significant health concern globally due to its aggressive nature and propensity for metastasis [1]. Among the various genetic alterations implicated in melanoma pathogenesis, the BRAFV600E mutation stands out as one of the most prevalent, occurring in approximately 50% of cases. This mutation, characterized by a valine-to-glutamic acid substitution at codon 600 of the BRAF gene, leads to constitutive activation of the mitogen-activated protein kinase (MAPK) signaling pathway, driving uncontrolled cell proliferation and tumor progression [2,3]. Additionally, mutations in the NRAS gene, such as NRAS Q61R, are found in about 15–20% of melanoma cases. These mutations also play a significant role in melanoma by activating the MAPK pathway through different mechanisms, contributing to the complexity and heterogeneity of the disease [4,5].

Despite significant advancements in melanoma treatment, including the advent of targeted therapies and immunotherapies, challenges remain in achieving durable responses and overcoming resistance mechanisms. Immunotherapy, particularly immune checkpoint inhibitors (ICIs) targeting programmed cell death protein 1 (PD-1) and cytotoxic T-lymphocyte-associated antigen 4 (CTLA-4), has revolutionized the treatment landscape of melanoma. However, a subset of patients does not respond to immunotherapy, while others experience initial responses followed by relapse, highlighting the complexity of immune evasion mechanisms in melanoma [6,7].

Understanding the underlying causes of treatment failure, including intrinsic tumor resistance and acquired resistance mechanisms, is crucial for optimizing treatment strategies and improving patient outcomes. Emerging evidence suggests that resistance to immunotherapy may arise from various factors, including tumor-intrinsic factors such as genetic alterations, tumor heterogeneity, and dysregulated signaling pathways, as well as extrinsic factors involving the tumor microenvironment (TME) and immune escape mechanisms [8,9,10,11].

In this context, immunogenic cell death (ICD) has emerged as a promising therapeutic strategy, triggering an immune response against tumor cells. ICD is characterized by the release of Damage-Associated Molecular Patterns (DAMPs), which activate the immune system [12]. Among various ICD-inducers, photodynamic therapy (PDT) has been investigated, which involves the administration of a photosensitizer (PS) followed by visible light irradiation, leading to the generation of reactive oxygen species (ROS) and localized oxidative stress. In our previous research, we demonstrated that the prodrug Me-ALA induces the production of endogenous PS protoporphyrin IX (PpIX) localized to the endoplasmic reticulum (ER) of murine melanoma cells, triggering ER-stress-mediated apoptotic cell death. PDT-treated melanoma cells also facilitated the maturation of monocyte-derived dendritic cells (DCs), enhancing co-stimulatory signals and chemotaxis toward tumors [13]. Understanding melanoma cell sensitivity to ICD and the underlying molecular mechanisms is crucial for developing effective immunotherapeutic approaches.

Here, our comprehensive analysis, integrating bioinformatics tools and experimental approaches, aimed to investigate the relationship between the BRAFV600E mutation and genomic alterations, immune landscape, and its impact on treatment response, in particular ICD, in melanoma. Our findings indicate a divergent sensitivity to specific ICD inducers, potentially linked to the BRAF mutation and its modulation of the interferon-1 (IFN-1) pathway. The IFN-1 signaling pathway, mediated by the regulation of interferon-stimulated genes (ISGs) in cancer, plays a crucial role in modulating the TME, regulating the anti-tumor immune response, and influencing therapy sensitivity [14,15]. Overall, our results reveal distinct genomic profiles and immune subtype dynamics associated with the BRAFV600E mutation in melanoma patients. Additionally, we provide insights into the complex interactions among BRAF signaling, ICD, and IFN-1 pathway activation, highlighting potential avenues for therapeutic intervention, immunomodulation, and enhancing responses to ICD in BRAF-mutated melanomas.

## 2. Materials and Methods

### 2.1. Cell Cultures

The BRAFV600E/NRAS WT human melanoma cell lines SK-MEL-28 and A375 were generously provided by Dr. Pérez Saez (IBYME) and Dr. Álvarez (IMIBIO-SL), respectively. The BRAF WT/NRAS Q61R human melanoma cell line SK-MEL-2 was generously provided by Dr. Álvarez (IMIBIO-SL). All cell lines were cultured according to ATCC protocols. Cells were maintained using DMEM (Modified Eagle medium of Dulbecco) (Gibco™, ThermoFischer Scientific, Buenos Aires, Argentina), supplemented with 10% (*v*/*v*) fetal bovine serum (FBS) (Internegocios, Buenos Aires, Argentina), and 1% antibiotic-antimycotic (Gibco™, ThermoFischer Scientific, Buenos Aires, Argentina) (10,000 units/mL penicillin, 10,000 μg/mL streptomycin and 25 μg/mL antifungal Fungizone), and were grown at 37 °C with 5% CO_2_ in a humidified incubator. The SK-MEL-2-IFN (IFN-1 pathway reporter) cell clone was generated following the subsequent instructions and cultured under identical conditions to the other cell lines.

### 2.2. Database Analysis

The TCGA PanCancer database, accessible through cBioPortal [16] or Xena [17], was utilized for the in silico analyses (accessed December 2023). Genomic and transcriptomic data were obtained from 363 skin cutaneous melanoma (SKCM) patients via cBioPortal, with evaluation of BRAF and NRAS gene mutation, mRNA expression of each gene of interest in diploid samples, and tumor mutation burden (TMB). Patients were categorized into two groups: (a) those with wild-type (WT) BRAF and those with the BRAFV600E mutation, and (b) those with wild-type (WT) NRAS and those with the NRAS Q61R mutation. mRNA expression of genes associated with immunogenic cell death (ICD), particularly DAMPs or their modulators, was analyzed in these groups. Additionally, data from 103 patients sourced from the Xena dataset were categorized based on BRAF WT protein form or BRAFV600E mutation, and immune subtypes [18] were assessed.

### 2.3. Cell Treatments

#### 2.3.1. Photodynamic Treatment:

Cell Lines Were Seeded into a 96-Well Plate (1 × 10^5^ cell/mL). The following day, cells were incubated with pro-drug 5-methylaminolevulinic acid (Me-ALA) (Sigma, Watham, MA, USA) (0.25, 0.50, 1, and 2 mM) in DMEM 1% FBS for 4 h to allow the endogenous generation of the protoporphyrin IX photosensitizer (PpIX). After incubation, tumor cells were irradiated at room temperature with a light dose of 1 J/cm^2^ (λ: 636 nm). The medium was then replaced with a fresh medium [13].

#### 2.3.2. Chemotherapeutic Treatments: 

Cell Lines Were Seeded into a 96-Well Plate (1 × 10^5^ cell/mL). The following day, cells were treated with doxorubicin (DXR) (Glenmark Life Sciences, Telangana, India) (1.5, 3, 6, and 12 μg/mL) or cisplatin (CP) (Deltapharma, Cairo, Egypt) (100, 200, 400, and 800 μg/mL) in DMEM 10% FBS for 24 h. 

#### 2.3.3. H151 (STING Inhibitor) Treatment and PDT: 

Cells Were Seeded into a 6-Well Plate (1 × 10^5^ cell/mL). The following day, cells were treated with 1 or 10 μM H151 or DMSO as a vehicle for DMEM 1% FBS for 4 h. A group of cells was simultaneously incubated with 1 mM Me-ALA, followed by the photodynamic treatment described above.

### 2.4. Cell Viability Assay

Cell viability was determined by the resazurin test using AlamarBlue™ (Invitrogen, Carlsbad, CA, USA). According to the manufacturer’s protocol, Alamar Blue dye was added to each well and incubated for 6 h protected from light. Fluorescence was read in Varioskan LUX (Thermo Fisher Scientific) at λem = 570 nm–λex = 590 nm. The lethal dose 50 (LD50), the concentration that induced death in 50% of the treated cells, was determined by preparing dose–response curves applying a non-linear regression to the data (GraphPad Prism 8).

### 2.5. Generation of IFN-1 Pathway Reporter Human Melanoma Cell Line

To generate the reporter cell line, SK-MEL-2 cells were seeded at a density of 3.5 × 10^4^ cells/mL in a 24-well plate. The following day, cells were transfected with the pMx2-eGFP plasmid containing the IFNα-inducing Mx2 promoter controlling GFP protein expression, using PEI 87 KDa (PolyAr87) in a reagent:DNA ratio of 2:1, as per the manufacturer’s instructions. The pMx2-eGFP plasmid was generously provided by Dr. María de los Milagros Bürgi (UNL) [19]. Stably transfected cells were selected by incubating them with G418 (600 μg/mL) for 21 days with a medium refresh every 2 days, followed by limiting dilution cloning. Selected cells were diluted to a final concentration of 5 cells/mL and seeded in 96-well plates with growth medium supplemented with G418. Once wells reached 100% confluence, they were gradually amplified to generate the SK-MEL-2-IFN cell clone. 

### 2.6. Flow Cytometry

After corresponding treatments, cells were trypsinized, resuspended in 1 mL of PBS, and centrifuged (1200 rpm, 4 min, 4 °C). For calreticulin staining, cells were incubated with the Alexa Fluor^®^ 488 Anti-Calreticulin antibody [EPR3924] (human) (AB196158, Abcam, Watham, MA, USA), according to the manufacturer’s instructions. The cellular sediment was stained with L/D reagent (Invitrogen™) for 10 min according to the manufacturer’s instructions for flow cytometry and then resuspended in FACS buffer (PBS 2% FBS). The samples were analyzed in the flow cytometer (Millipore, Darmstadt, Germany, Guava Easycyte 6 2L) to quantify GFP expression on live cells (L/D negative). For each sample, 10,000–15,000 events were acquired in the analyzed region. Data were analyzed with FlowJo vx 0.7 software (Tree Star Inc., Ashland, OR, USA).

### 2.7. Statistical Analysis

The experimental data are representative of at least three independent experiments and expressed as the means ± SEM and each dot in the graphs represents a biological replicate. Statistical data are informed in the corresponding figure legend. GraphPad Prism 8 (version 8, GraphPad Software, San Diego, CA, USA) software was utilized to carry out the work.

## 3. Results

### 3.1. Association of BRAFV600E Mutation with Genomic and Immune Landscape Alterations in Melanoma Patients

To investigate the impact of BRAFV600E mutation, known for its high prevalence in melanoma [2], on immunogenomic and immune microenvironment in melanoma patients (SKCM), we first conducted an in silico analysis using data from The Cancer Genome Atlas (TCGA) accessed through cBioPortal and Xena platforms. The cBioPortal dataset, comprising TCGA PanCancer data, included 363 patients, with 48.21% exhibiting the wild-type (WT) form of the BRAF protein and 34.44% harboring the BRAFV600E mutation, which was the most prevalent mutation (Figure 1A). We identified that the BRAFV600E mutation in melanoma patients is significantly associated with a lower Tumor Mutation Burden (TMB) (Figure 1B). TMB, reflecting the total number of non-synonymous mutations per megabase of the genome, serves as a metric for genomic instability and potential neoantigen formation [20,21]. Since activating mutations in RAS oncogenes are found in a third of all human cancers and NRAS mutations are found in 15–20% of melanomas [22], we decided to study this as well. Analyzing the same database, we observed that the NRAS Q61R mutation was the most prevalent in melanoma patients (12.4%) (Appendix A), but unlike the BRAFV600E mutation, it did not alter the TMB (Appendix A). Analysis of the Xena dataset, also derived from TCGA PanCancer, included 103 patients, among whom 46.60% had the WT BRAF gene form, and 47.67% had the BRAFV600E mutation, confirming this mutation as the most prevalent (Figure 1C). A recent meta-analysis thoroughly examined the immune TME using transcriptomic data, uncovering six distinct immune subtypes across different tumor types [18]. Here, the investigation into these immune subtypes unveiled the impact of the BRAFV600E mutation on the profile within the melanoma immune landscape (Figure 1D). Specifically, in BRAFV600E-mutant tumors, there was an abundance of an “Immune C1: Wound healing” profile observed, along with an increased prevalence of other subtypes associated with immune system dysfunction, such as “Immune C4: Lymphocyte Depleted” and “Immune C6: TGF-beta Dominant”. Conversely, subtypes indicative of a favorable immune response or infiltration, such as “Immune C2: IFN-gamma Dominant and Immune C3: Inflammatory types”, showed decreased prevalence in melanoma patients carrying the BRAFV600E mutation. Our findings underscore that the presence of the BRAFV600E mutation is linked to a reduced TMB and an overall immunosuppressive TME.

### 3.2. Sensitivity of Cell Lines Harboring Different Genomic Profiles to ICD and Non-ICD Inducer and Their Impact on ICD-Associated Gene Expression Profiles in Melanoma

Given the impact of the genomic profile on the immune landscape, particularly the BRAFV600E mutation (Figure 1), we aimed to evaluate the impact of this mutation on sensitivity to different antitumor approaches. We focused on those that have been reported to enhance the induction of immunogenic cell death (ICD), a form of cell death that elicits an immune response against tumor cells [12]. To address this, we conducted in vitro cytotoxicity assays using melanoma cell lines harboring either BRAF wild-type (WT) (SK-MEL-2) or BRAFV600E mutations (SK-MEL-28 and A375). These models have been widely used by other researchers to represent this genetic context in melanoma studies [23,24,25]. It is important to note that SK-MEL-2 cells, although BRAF wild-type, carry the NRAS Q61R mutation, which activates the NRAS pathway. In contrast, SK-MEL-28 and A375 are NRAS wild-type. However, data from other studies demonstrate that ERK1/2 is activated in SK-MEL-28 and A375 cells, but not in SK-MEL-2 cells, indicating differential pathway activation despite the NRAS mutation [22,26]. This supports the use of SK-MEL-2 cells as a control model in our study and underscores their relevance in investigating the effects specific to the BRAF mutation. These cell lines were exposed to various ICD inducers, including doxorubicin [27,28,29,30] and Me-ALA-based photodynamic therapy (PDT) [13], as well as a non-ICD inducer, cisplatin [31,32]. Across all treatments, dose–response curves demonstrated that the decrease in cell viability was directly proportional to the concentration of the respective drug used (Figure 2A–C). Notably, substantial distinctions were observed specifically in the context of PDT, as BRAFV600E mutant cell lines showed increased sensitivity compared to BRAF WT SK-MEL-2 (Figure 2A). The absorption of light by melanin would not explain the observed resistance to PDT in our model. The LED wavelength used (636 nm) ensures that the absorption by PpIX does not interfere with melanin absorption [33]. Furthermore, none of the human cell lines (SK-MEL-2, SK-MEL-28, and A375) produced detectable melanin, unlike the murine B16-F10 melanoma line known for melanin production (Appendix A). These results negate the hypothesis that melanin could shield the cells from effective PDT irradiation.

Conversely, while A375 exhibited the highest resistance to doxorubicin (Figure 2B) and SK-MEL-28 displayed the highest resistance to cisplatin (Figure 2C), no notable distinctions were observed between the remaining cell lines in both instances, with one carrying the BRAF wild-type (WT) and the other bearing the BRAFV600E mutation (Table 1).

To further explore the impact of BRAFV600E mutation on the expression of specific genes associated with ICD, particularly DAMPs, we analyzed data from the cBioPortal database (Figure 1A). Surprisingly, our analysis revealed an upregulation in the expression of genes associated with the IFN-1 pathway [34]: IFNAR1, IFNAR2, and CXCL10 in BRAFV600E-mutated tumors (Figure 2D). Additionally, we observed a decrease in the expression of ectonucleotidase CD39 [35], which plays a role in inhibiting and reversing the pro-inflammatory actions of ATP (Figure 2E). The expression of none of these genes changed in the presence of the NRAS Q61R mutation (Appendix A).

Our findings suggest a potential link between the BRAFV600E mutation on the responsiveness to the ICD inducer PDT. These results raise the question of whether this differential sensitivity might be associated with the distinct profile observed in the IFN-1 pathway, which appears to be more pronounced in BRAFV600E-mutated tumors.

### 3.3. Modulation of IFN-1 Pathway Activity in Human Melanoma Cells by PDT-Induced ICD

The activation of the IFN-1 pathway by ICD was first reported using anthracyclines as an ICD inducer [36]. Additionally, our group observed an upregulation of IFN-α, IFN-β, and certain pro-inflammatory ISG expressions following Me-ALA-based PDT in a murine melanoma model linked to dendritic cell maturation [13]. In the present study, we aimed to investigate whether ICD could promote the upregulation of this pathway in a human melanoma model. To address this question, we utilized SK-MEL-2 cells transfected with the MX2-EGFP plasmid, which contains the MX2 promoter, a well-known ISG, controlling the expression of GFP protein [19]. After selection with geneticin, we generated the SK-MEL-2-IFN clone, which serves as a reporter for IFN-1 pathway activity. Subsequently, SK-MEL-2-IFN cells were exposed to cytotoxic doses of various ICD inducers, including doxorubicin and Me-ALA-based photodynamic therapy (PDT), as well as the non-ICD inducer cisplatin. GFP expression, indicative of IFN-1 pathway activity, was evaluated using flow cytometry. PDT emerged as the sole ICD inducer capable of robustly upregulating this pathway, as evidenced by a notable increase in the percentage of GFP+ cells, exhibiting IFN-1 pathway activation (Figure 3A). This upregulation of the IFN-1 pathway was accompanied by an increase in CRT expression on the cell surface (Appendix A). CRT is considered a pivotal marker of ICD [37]. Unlike previous reports by other researchers [38], treatment with doxorubicin did not show modulation of the IFN-1 pathway in our experimental setup (Figure 3B). In contrast, the non-ICD inducer cisplatin even led to a significant downregulation of the IFN-1 pathway (Figure 3C). These findings highlight the varied impacts of distinct cell death inducers on the IFN-1 pathway, emphasizing the importance of delving deeper into the mechanisms that govern these responses and their subsequent implications. Collectively, our results suggest that the BRAFV600E mutation in melanoma, potentially by upregulating the IFNAR1/2, renders distinct susceptibility to antitumor therapies that induce IFN-1 pathway, such as the ICD-inducer PDT.

### 3.4. Inhibition of cGAS-STING Signaling Reverses PDT-Induced Upregulation of IFN-1 Pathway Activity Le Nano

Many anti-tumor therapeutics exert their effects through cytotoxic mechanisms, often involving the destruction of chromosomal DNA [39,40,41,42]. The cGAS-STING signaling pathway serves as a crucial cytoplasmic DNA-sensing pathway, playing a pivotal role in regulating immune responses to cancer, infections, and autoimmune diseases by triggering the production of IFN-1 which modulates immune regulation [43]. Building on the preceding findings, we sought to explore whether cGAS-STING signaling might be involved in regulating IFN-1 triggered by PDT. To investigate this hypothesis, we employed the STING inhibitor H-151 [44,45,46]. Cells were pre-incubated with the inhibitor prior to PDT treatment. At a concentration of 10 µM, the STING inhibitor was capable of reversing the upregulation of the GFP+ cell population indicative of an active IFN-1 pathway, as well as the transcriptional activity associated with this pathway. Notably, this reversal effect was observed only under conditions of PDT treatment and not in basal conditions (Figure 4). The cGAS-STING pathway inhibitor did not alter the PDT-induced surface exposure of CRT (Appendix A). Overall, our findings suggest a specific interaction between PDT-induced cellular processes and the cGAS-STING signaling pathway, wherein the presence of the STING inhibitor selectively interfered with the downstream activation of IFN-1 pathway triggered by PDT-induced cell death mechanisms. Our study highlights the context-specific nature of cGAS-STING signaling modulation and the intricate interplay between PDT-mediated cellular responses and immune signaling pathways. 

## 4. Discussion

Melanoma represents a paradigmatic example of the intricate interplay between the immune system and cancer progression. Despite being one of the most immunogenic tumors due to its high genomic mutational burden [4,21], melanoma paradoxically evades immune surveillance through diverse mechanisms [8,9,10,11,47]. This dynamic interaction between melanoma cells and the immune system significantly influences disease progression and treatment outcomes. Thus, comprehending this interplay is crucial for developing effective immunotherapeutic strategies against this aggressive malignancy. Furthermore, the BRAFV600E mutation, found in about 50% of melanomas, complicates the melanoma landscape by driving oncogenic signaling pathways, leading to increased cell proliferation, survival, and metastasis [3,48]. BRAF mutation status guides targeted therapy choices, while immunological features of the TME influence the efficacy of immunotherapeutic approaches in melanoma. Integrating molecular and immune profiling is essential for optimizing treatment outcomes [49,50]. Our study provides comprehensive insights into this complex interplay in melanoma.

Through in silico analyses, we revealed distinct genomic and immune landscape alterations associated with the BRAFV600E mutation in melanoma. Consistent with previous reports [2], we found a heightened frequency of this mutation among melanoma patients. Importantly, this mutation significantly correlated with a decreased TMB, potentially impairing neoantigens formation and hindering immune recognition and response. These findings are in line with studies on other cancers, indicating that probably in the presence of oncogene-driven mutations, including BRAFV600E, the contribution of additional mutations is dispensable in sustaining cancer cell survival and proliferation. A high TMB has been associated with improved clinical outcomes after treatment with immune checkpoint inhibitors (ICIs) [51,52,53,54,55,56,57]. Melanoma, characterized by a high TMB, underscores the intricate relationship between genomic alterations and immune evasion mechanisms [20,21].

It is widely recognized that BRAFV600E in melanoma can act as an immunogenic peptide when presented on MHC-II by CD4+ T-cells. Nonetheless, this mutation is linked to elevated expression of immunosuppressive factors and decreased antigen presentation by MHC-I. The application of BRAF inhibitors has shown promise in reversing tumor-associated immunosuppressive signals [58]. However, it is worth noting that there is limited literature available on this topic, highlighting the need for further research. In a recent meta-analysis, significant efforts were made to comprehensively characterize the immune TME across 33 different cancers as analyzed by TCGA. Through this integrated approach, researchers delineated and described six distinct immune subtypes present across various tumor types, highlighting their potential therapeutic and prognostic relevance in cancer management [18]. Notably, our investigation into immune subtypes uncovered substantial alterations within the immune microenvironment of melanoma tumors bearing the BRAFV600E mutation. Specifically, we observed an exacerbation of immune subtype profiles associated with tumor immune evasion, including the “Immune C1: Wound healing profile”, characterized by elevated expression of angiogenic genes and a high proliferation rate. Additionally, we noted an increase in subtypes characterized by the loss of immune function, such as “Immune C4: Lymphocyte Depleted” and “Immune C6: TGF-beta Dominant”, which displayed a more prominent macrophage signature and a high TGF-beta signature, respectively. Conversely, subtypes indicative of a favorable immune response, such as “Immune C2: IFN-gamma Dominant” and “Immune C3: Inflammatory”, exhibited decreased prevalence in BRAFV600E-mutated melanoma patients. These findings underscore the immunosuppressive nature of BRAFV600E-mutant tumors compared to BRAF wild-type ones.

In the context of advancing immunotherapies, ICD has emerged as a promising strategy to enhance the immunogenicity of dying cancer cells, potentially boosting the effectiveness of immunotherapeutic interventions [59,60,61]. In this study, we aimed to study the possible relationship between the BRAFV600E mutation and the response to ICD induction, seeking to identify strategies for improving immunostimulant therapies against melanoma. To this end, we explored the responsiveness of BRAFV600E-mutated cells to ICD inducers, including doxorubicin [27,28,29,30] and PDT using Me-ALA as a prodrug [13]. Indeed, our group was a pioneer in describing Me-ALA-based photodynamic therapy (PDT) as an inducer of ICD [13]. Additionally, we included cisplatin as a control, a conventional chemotherapeutic agent lacking ICD-inducing properties [31,32]. Our findings suggest that tumors harboring the BRAFV600E mutation may exhibit increased susceptibility to PDT-induced cell death. This phenomenon underscores the complex interplay between oncogenic mutations and therapeutic responses in melanoma. The specific molecular alterations driven by the BRAFV600E mutation that likely contribute to the observed differential sensitivity need to be explored. Our preliminary analysis of transcriptomic data concerning DAMPs revealed intriguing alterations associated with the BRAFV600E mutation, particularly a downregulation in genes related to ATP metabolism (CD39) and upregulation of the type 1 interferon (IFN-1) pathway (IFNAR1, IFNAR2, CXCL10). CD39, along with CD73, converts extracellular ATP (a well-known DAMP) to adenosine, which inhibits T-cell effector functions via the adenosine receptor A2A [62]. Further investigation into the modulation of the cytokine CXCL10 in melanoma cell lines corroborated our findings. Immunohistochemistry (IHC) analysis demonstrated that BRAFV600E-mutated cell lines (A375 and SK-MEL-28) exhibited higher intracellular expression of CXCL10 compared to the wild-type BRAF cell line (SK-MEL-2). Interestingly, this increased intracellular expression did not correlate with the secretion levels of CXCL10. These results align with our transcriptomic data, reinforcing the notion that the BRAFV600E mutation influences the expression of key ICD-related cytokines. This observed discrepancy between intracellular expression and secretion of CXCL10 suggests a complex regulatory mechanism that warrants further investigation [63].

The IFN-1 pathway plays a critical role in mediating anti-tumor immunity and is involved in the response to various cancer therapies [64,65,66]. One intriguing aspect highlighted by our results is the potential association between differential sensitivity to ICD and the distinct profile observed in the IFN-1 pathway in BRAFV600E-mutated tumors. 

Interferon (IFN)-α2b, as the first approved immunotherapy for melanoma, has historically demonstrated significant benefits in improving both relapse-free survival and overall survival. While no longer a first-line treatment, ongoing research investigates its potential as an adjuvant in combination therapies, highlighting its continued relevance in enhancing the efficacy of other immunotherapies for melanoma patients [67]. The IFN-1 pathway was recently identified as a DAMP involved in ICD [38], which differs from constitutively expressed cDAMPs like calreticulin, ATP, and HMGB1. As an inducible DAMP [68], IFN-1 activation post-ICD serves as a crucial mediator of anti-tumor immunity, facilitating immune effector cell recruitment, antigen presentation enhancement, and durable immune memory generation. Triggered by various stimuli, including viral infections and nucleic ligands, IFN-1 production involves PRRs and cytoplasmic sensors like cGAS, activating transcription factors such as IRFs and NF-κB, culminating in IFN-α and IFN-β production and downstream signaling via IFNAR1/2 receptors [34,36,69,70].

The establishment of the SK-MEL-2-IFN reporter cell line, specifically engineered to express a fluorescent reporter under the control of the IFN-1 pathway [19], played a pivotal role in our investigation into the impact of ICD on the activation of this crucial signaling cascade in melanoma. The establishment of this experimental framework allowed us to monitor and quantify the basal activity of the IFN-1 pathway in SK-MEL-2 melanoma cells. It is important to note the basal activity of the IFN-1 pathway observed in these cells is likely associated with the basal activity of IFN-1 exhibited by tumor cells, which can either promote cytotoxicity or confer pro-survival advantages depending on the strength and duration of the response, thereby impacting cancer therapy efficacy [15].

Here, we observed a notable upregulation of the IFN-1 pathway specifically in response to PDT, aligning with our previous findings in a murine melanoma model [13]. Previously, we demonstrated the induction of phosphorylated IRF3 and upregulation of key ISGs, including the cGAS receptor and phosphorylated STAT1, during PDT, suggesting potential autocrine stimulation. Our ongoing investigation aims to elucidate cGAS-STING signaling mechanisms, involving cytoplasmic DNA recognition by cGAS and STING activation [71]. Importantly, inhibiting cGAS-STING with H151 reverses PDT-induced IFN-1 pathway upregulation. Previous studies demonstrated that radiotherapy and certain chemotherapeutic drugs induce cytotoxicity in cancer cells, releasing DNA fragments that activate cGAS-STING, prompting IFN-1 production and immune responses [39,40,41,42]. Our findings expand our understanding of this pathway’s role in regulating PDT-induced IFN-1 activity in melanoma, warranting further investigation into the responsible DNA-associated ligand. Emerging studies highlight the synergy between STING agonists and ICIs, enhancing anti-tumor immunity. The cGAS-STING pathway plays a crucial role in innate immune recognition of immunogenic tumors, facilitating APC maturation, cytokine secretion, and CD8+ T cell development targeting tumor-specific antigens. Activation of this pathway reshapes the TME, boosting the anti-tumor immune response and promising new therapeutic approaches for melanoma and other cancers [43]. Given the established curative and synergistic effects observed in preclinical studies with various therapeutic modalities [14], it is plausible that innovative strategies incorporating IFN-1 system activation as part of combination therapies will lead to even better response rates and survival outcomes.

The observation that BRAFV600E tumors are associated with a more tolerogenic TME while cells harboring the same mutation are more susceptible to ICD via IFN-1 is intriguing and warrants further discussion. The paradox may be reconciled by considering the distinct temporal and spatial effects of BRAFV600E on the TME and the direct impact of IFN-1 on tumor cells. While the mutation initially promotes a tolerogenic TME, the introduction of IFN-1 by ICD would disrupt this environment. This can be explained by the role of IFN-1 in ICD, which also involves the release of other DAMPs such as HMGB1, ATP, and calreticulin exposure. These DAMPs enhance the recruitment and activation of DCs and effector T cells, thereby promoting a robust anti-tumor immune response [59]. Understanding this paradox has significant clinical implications. Therapies that can induce ICD may be particularly effective in BRAFV600E tumors by converting the tolerogenic TME into an immunogenic one.

## 5. Conclusions

Among the ICD inducers evaluated, PDT emerged as an inducer capable of robustly activating the IFN-1 pathway in a specific cellular context. The upregulation of IFNAR1/2 associated with the BRAFV600E mutation might be linked to the enhanced susceptibility of tumor cells to PDT, potentially rendering BRAFV600E-mutated cells more responsive to stimuli that activate the IFN-1 pathway. While these findings provide valuable insights, it is important to acknowledge the limitations of our study, particularly due to the use of non-isogenic models. However, this approach allowed us to utilize the complete genomic landscape to describe differences across various melanoma cell lines. Our observations highlight that the need for further investigation is warranted to validate and elucidate the mechanistic foundations of this phenomenon. Exploring the broader implications of the IFN-1 pathway in the context of different cell death inducers may offer deeper insights into the complex interplay between oncogenic mutations, immune responses, and therapeutic strategies in melanoma treatment.

## Figures and Tables

**Figure 1 cancers-16-02568-f001:**
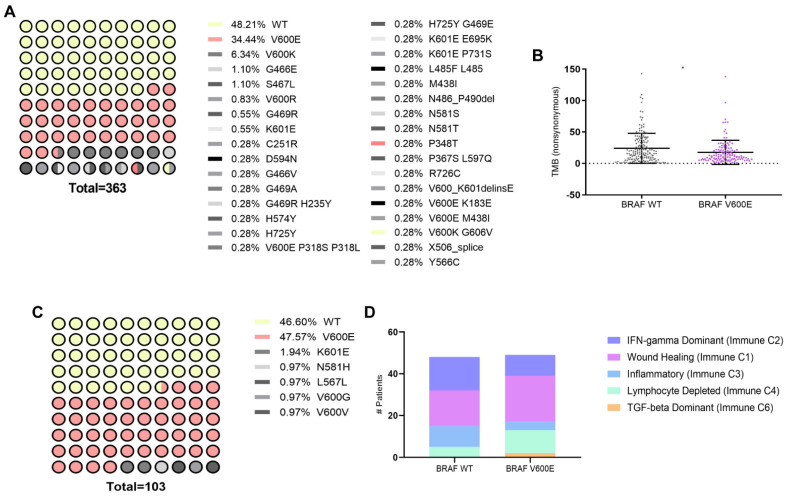
Genetic and immunological landscape of BRAF mutations in SKCM. (**A**) Distribution of mutation frequencies in BRAF among skin cutaneous melanoma (SKCM) patients, sourced from cBioPortal (TCGA Pan-Cancer Atlas dataset). (**B**) Dot plot showing tumor mutational burden (TMB) comparison between BRAF wild type (WT) (*n* = 175) and BRAF V600E (*n* = 125) mutations in SKCM, quantified by the number of proteins carrying non-synonymous mutations. Statistical analysis was conducted using the Student *t*-test for unmatched samples: * *p* < 0.05. (**C**) Mutation frequency distribution in BRAF among SKCM patients, sourced from Xena (TCGA Pan-Cancer Atlas dataset). (**D**) Bar plot showing the classification of immune subtypes in SKCM patients based on BRAF WT (*n* = 48) and BRAF V600E (*n* = 49) mutations.

**Figure 2 cancers-16-02568-f002:**
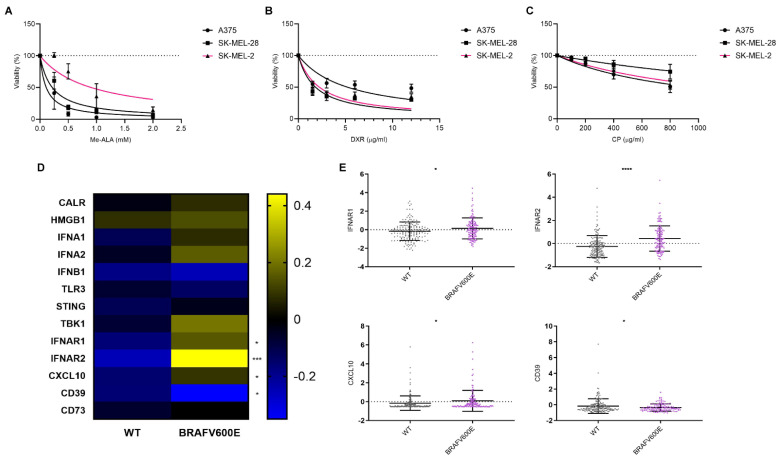
Response variation of melanoma cell lines to immunogenic cell death inducers and associated gene expression profiles in the context of BRAF mutation. Melanoma cell lines with BRAF wild-type (SK-MEL-2, pink line) or BRAF V600E mutations (SK-MEL-28 and A375, black lines) were exposed to increasing doses of the ICD inducers: (**A**) photodynamic therapy (PDT): cells were first incubated with increasing doses of the pro-drug Me-ALA (0–2 mM) for 4 h, followed by irradiation with a light dose of 1 J/cm^2^ (λ: 636 nm); (**B**) doxorubicin (DXR): cells were incubated to increasing doses of the chemotherapeutic (0–12 µg/mL) for 24 h; (**C**) cisplatin (CP): cells were incubated to increasing doses of the chemotherapeutic (0–800 µg/mL) for 24 h. Cell viability was assessed 24 h after treatment using the resazurin assay and expressed as a percentage of non-treated cells (100% represented by the dotted line). Dose–response curves were generated using non-linear regression analysis (GraphPad Prism). (**D**) Heatmap illustrating mRNA expression levels of genes associated with immunogenic cell death (ICD), particularly DAMPs or their modulators, in BRAF wild-type (*n* = 175) and BRAF V600E (*n* = 125) melanoma samples sourced from cBioPortal (TCGA Pan-Cancer Atlas dataset). Statistical analysis was conducted using the Student *t*-test for unmatched samples: *** *p* < 0.0001. * *p* < 0.05. (**E**) Dot plots showing mRNA expression levels of genes with statistically significant differences in BRAF V600E samples compared to BRAF wild-type samples. Statistical analysis was conducted using the Student *t*-test for unmatched samples: **** *p* < 0.0001. * *p* < 0.05.

**Figure 3 cancers-16-02568-f003:**
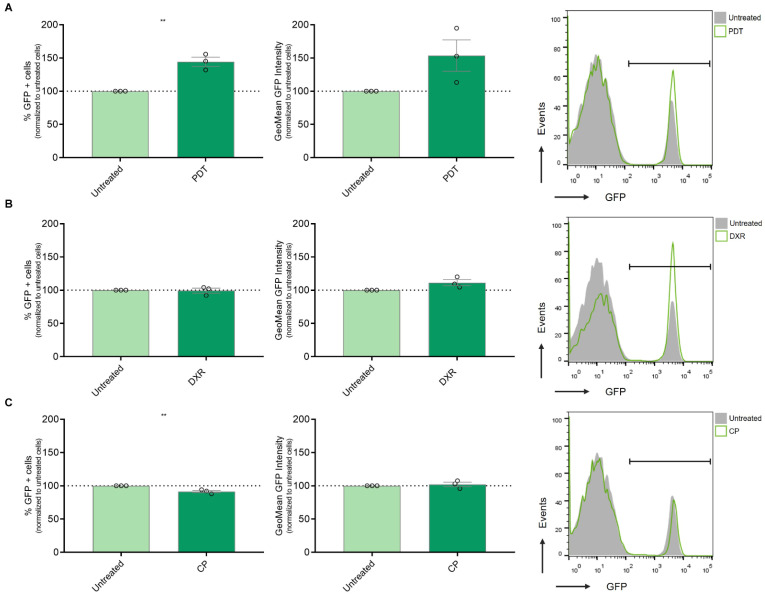
Assessment of IFN-1 Pathway Activation in Melanoma Cells Exposed to Immunogenic and Non-Immunogenic Cell Death Inducers. (**A**) Gene reporter assay was performed using the IFN-1 pathway reporter cell line SK-MEL-2-IFN exposed to the non-ICD inducer cisplatin (CP) (800 µg/mL) for 24 h. GFP expression on live cells was quantified from flow cytometry data using FlowJo. Data is presented as the percentage of GFP-positive (GFP+) cells representing IFN-1 pathway activation (left), with analysis of the level of IFN-1 activation in GFP+ positive cells performed from GeoMean data (center), all normalized to values corresponding to untreated cells (100% represented by the dotted line). Statistical analysis was conducted using the Student *t*-test for unmatched samples: ** *p* < 0.01. Representative histograms are shown (right). (**B**) Gene reporter assay was performed using the IFN-1 pathway reporter cell line SK-MEL-2-IFN exposed to the immunogenic cell death (ICD) inducer doxorubicin (DXR) (6 µg/mL) for 24 h. GFP expression on live cells was quantified from flow cytometry data using FlowJo. Data is presented as the percentage of GFP-positive (GFP+) cells representing IFN-1 pathway activation (left), with analysis of the level of IFN-1 activation in GFP+ positive cells performed from GeoMean data (center), all normalized to values corresponding to untreated cells (100% represented by the dotted line). Statistical analysis was conducted using the Student *t*-test for unmatched samples: absence of asterisks indicates a statistically non-significant difference. Representative histograms are shown (right). (**C**) Gene reporter assay was performed using the IFN-1 pathway reporter cell line SK-MEL-2-IFN exposed to the ICD inducer photodynamic therapy (PDT). Cells were first incubated with the pro-drug Me-ALA (1 mM) for 4 h, followed by irradiation with a light dose of 1 J/cm^2^ (λ: 636 nm). GFP expression on live cells was quantified from flow cytometry data using FlowJo. Data is presented as the percentage of GFP-positive (GFP+) cells representing IFN-1 pathway activation (left), with analysis of the level of IFN-1 activation in GFP+ positive cells performed from GeoMean data (center), all normalized to values corresponding to untreated cells (100% represented by the dotted line). Statistical analysis was conducted using the Student *t*-test for unmatched samples: ** *p* < 0.01. Representative histograms are shown (right).

**Figure 4 cancers-16-02568-f004:**
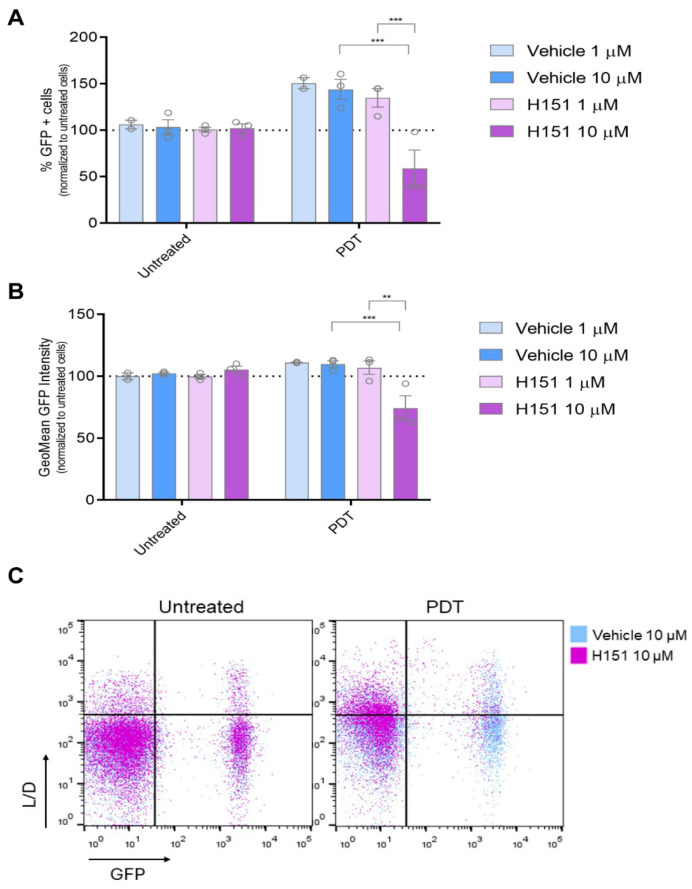
Modulation of IFN-1 Pathway Activation by PDT mediated by cGAS-STING signaling. Gene reporter assay was performed using the IFN-1 pathway reporter cell line SK-MEL-2-IFN exposed to the immunogenic cell death (ICD) inducer photodynamic therapy (PDT) in the presence or absence of the STING inhibitor H151 (1 mM and 10 mM). Cells were first incubated with the pro-drug Me-ALA (1 mM) for 4 h with or without H151 or vehicle (DMSO), followed by irradiation with a light dose of 1 J/cm^2^ (λ: 636 nm). GFP expression on live cells (L/D negative), indicative of IFN-1 pathway activation, was quantified from flow cytometry data using FlowJo. The data is presented as the percentage of GFP-positive (GFP+) cells representing IFN-1 pathway activation (**A**), with analysis of the level of IFN-1 activation in GFP+ positive cells performed from GeoMean data (**B**), all normalized to values corresponding to untreated cells (100% represented by the dotted line). Statistical analysis was conducted using two-way ANOVA with Tukey’s post hoc test: ** *p* < 0.01, *** *p* < 0.001. Representative dot plots are shown (**C**).

**Table 1 cancers-16-02568-t001:** Statistical Comparison of IC50 Values for Melanoma Cell Lines in Response to Different Treatments. Table 1 displays the *p* values obtained from statistical comparisons of the IC50 (inhibitory concentration 50%) values derived from nonlinear regression analysis of dose–response curves for each pair of cell lines in response to various treatments (PDT: photodynamic therapy, DXR: doxorubicin, CP: cisplatin). An extra sum-of-squares F test was performed to evaluate differences in best-fit parameters (IC50) among cell lines within each treatment (GraphPad Prism). Statistically significant *p* values (<0.05) are highlighted in blue, while non-significant ones are highlighted in yellow. The grey cells correspond to intersections in the table between same cell lines.

PDT			
	SK-MEL-2	A375	SK-MEL-28
SK-MEL-2		0.0001	0.0001
A375	0.0001		0.87
SK-MEL-28	0.0001	0.87	
DXR			
	SK-MEL-2	A375	SK-MEL-28
SK-MEL-2		0.0072	0.4364
A375	0.0072		0.005
SK-MEL-28	0.4364	0.005	
CP			
	SK-MEL-2	A375	SK-MEL-28
SK-MEL-2		0.2866	0.003
A375	0.2866		0.0004
SK-MEL-28	0.003	0.0004	

## Data Availability

The original contributions presented in the study are included in the article/Appendix A, further inquiries can be directed to the corresponding author.

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
