# Peer review of "Impact of Genomic Mutation on Melanoma Immune Microenvironment and IFN-1 Pathway-Driven Therapeutic Responses"

_cancers, 2024, doi:10.3390/cancers16142568_

Round 1

Reviewer 1 Report (Previous Reviewer 1)

Comments and Suggestions for Authors

in this revised manuscript, the authors did a good job answering my concerns.

Author Response

We are pleased to have met your expectations and greatly appreciate your positive feedback.

Reviewer 2 Report (Previous Reviewer 3)

Comments and Suggestions for Authors

I have considered carefully the rebuttal letter presented by the authors, as well as the amendments/additions introduced in the revised version of manuscript ID: cancers-3096442. I am still quite skeptical about the design and significance of the study. In general terms, I do not feel that the new wording acknowledges clearly enough the limitations of the study, or states explicitly the preliminary and speculative nature of the conclusions. Concerning specific points, in the reply to comment 2, an analysis of ERK pathway status is still missing (other than the inclusion of some references). Moreover, NRAS mutations have the potential to trigger ERK-independent survival pathways (eg PI3K/AKT) that may not be activated downstream of mutant BRAF, thus making NRAS- and BRAF-mutated melanoma cell lines hardly comparable. Also importantly, the fact that the cGAS-STING pathway, highly relevant for this study, is deficient in SKMEL28 and A375 cells is a serious concern that strengthens the limitations of the use of non-isogenic cellular models. Finally, the new data in the rebuttal letter concerning LU1205 cells are not too convincing. Therefore, in my opinion the manuscript still needs revision further emphasizing its limitations and including the new materials mentioned in the reply to points 2 and 5 of the rebuttal.

Author Response

Dear Reviewer 2:
We appreciate your detailed feedback and the opportunity to further improve our manuscript. We understand and acknowledge that our experimental design has its limitations, particularly the use of non-isogenic cell lines. However, our design was based on a thorough analysis of the existing literature, where many researchers use SK-MEL-2 as BRAF WT and SK-MEL-28 and A375 as BRAF-mutated cell lines. Based on these precedents, we constructed our experimental platform. To provide greater clarity and justification for our design choices, we have included additional references in the manuscript (see lines 227-228, ref 23-25).

We have also revised certain statements in the section 3.2 (lines 269-272) and conclusion (lines 542-547) to avoid overstatements. We explicitly emphasize that this study serves as an initial comparison of different cell lines and that confirmation would require further studies with isogenic models. We have also expanded the introduction to discuss in more detail the variation in the genomic landscape observed in melanoma, including NRAS mutation (lines 52-55).

Despite these limitations, we believe that our study meets its primary objective of comparing different genomic landscapes and analyzing how cellular characteristics might influence treatment response.

In response to your suggestions and to highlight the importance of our observations in PDT, we have included detailed information about melanin production in the cells and light irradiation (lines 241-246) and added a photograph of the cell pellets in the new supplementary figure 2.

Regarding the LU1205 cell line, we acknowledge that although it is reported as BRAFV600E, it is not as commonly used in the literature, at least to our knowledge. Nonetheless, if the editor deems it necessary, we are willing to include the new viability curves. However, it is important to note that these additional data do not lead to different conclusions.

We hope these revisions address your concerns and improve the clarity and impact of our manuscript.

Thank you for your constructive feedback.

This manuscript is a resubmission of an earlier submission. The following is a list of the peer review reports and author responses from that submission.

Round 1

Reviewer 1 Report

Comments and Suggestions for Authors

In the work by Mentucci et al, the authors demonstrated that melanomas with the BRAFV600E mutation tend to correlate with lower tumor mutation burden and with immunosuppression  gene signatures, suggesting these tumors would respond less to immunotherapy. They also show that  BRAFV600E melanoma cells were sensitive to ICD induced by photodynamic therapy, for example, and that this response was partially linked to IFN-1 signaling activation via cGAS-STING. Overall, this is a well thought out and well written manuscript that warrants publication in Cancers, provided a few points are clarified:

1)        Fig. 2E. quantification of cytokines: are these data from the TCGA? What about your cells?

2)        You mention Doxorubicin did not modulate ICD in your cells. Before concluding this is a general case, additional lines should be added in the analysis. 2 BRAFmut and 1RASmut line are not enough to draw clear cut conclusions. Adding more BRAFmut and RASmut  lines as well as perhaps WT/WT lines would strengthen the data overall.

3)        The same goes for testing IFN. What about other DAMPs? It is possible others are produced and adding these data would certainly strengthen this work.

4)        Blockade of STING reduces IFN-1 pathway (reporter data). Does the inhibitor also rescue ICD?  

5)         It is very interesting that BRAFV600E tumors associate with a more tolerogenic TME and that cells with the same mutation appear to be more susceptible to ICD via IFN-1, which should induce an anti-tumor immune response. This point should be further discussed as it is almost counterintuitive and definitely an interesting point of discussion.

Reviewer 2 Report

Comments and Suggestions for Authors

In this paper, Mentucci FM et al aimed to assess the association between BRAFV600E mutation, immune subtype dynamics, tumor mutational burden, and IFN-1 pathway in melanoma, by using bioinformatics approaches. Next, the authors also investigated the sensitivity of melanoma cell lines to ICD inducers, depending on the BRAFV600E mutation. The topic is potentially relevant, and the comparison of the six immune subtypes in BRAFwt vs BRAFV600E melanoma patients is interesting.

Despite that, this paper shows some concerns, especially about the experimental plan of in vitro experiments and the novelty of the results.

- The use of two BRAF V600E and one BRAFwt cell line is insufficient to conclude that BRAFV600E cells are more sensitive to PDT than BRAFwt ones (Fig. 1A-C). A panel of BRAF mut and wt cells is required for this aim.

- The effects the authors reported on the cells do not seem generalizable to ICD activation but rather limited to PDT treatment alone. How do the authors explain that doxorubicin is not capable of activating the IFN-1 pathway?

- Whether this paper is focused on the evaluation of the role of BRAFV600E, why did the authors decide to assess the IFN1 activation in the BRAFwt melanoma cell line (SKMEL2) (Fig.3-4)?

-Finally, referring to FIG 3-4, the authors seem to use an alternative approach to demonstrate that they have already demonstrated in their previous paper (PDT activates IFN-1 pathway through cGAS/STING, ref 11). What is the novelty of this paper?

Comments on the Quality of English Language

The quality of English language is satisfactory

Reviewer 3 Report

Comments and Suggestions for Authors

Manuscript cancers-2919283 by Mentucci et al describes a potentially interesting study of the role of the frequent V600E BRAF mutation on modulation of the immune microenvironment and IFN-1 pathway activity in skin melanoma (SKCM). The authors report a combination of in silico analyses of the SKCM dataset of TCGA and in vitro studies using 3 different melanoma cell lines, one of them BRAF-wildtype (BRAFwt, SK-MEL-2) and the other 2 bearing the BRAF-V600E driver mutation (SK-MEL-28 and A375). Concerning in silico analyses, the authors show that the V600E BRAF mutation correlates with a lower mutation burden and with immune subtypes suggesting  immunosuppression. The authors also compared in vitro the response of the 3 cell lines to the immunogenic cell death (ICD) inducers doxorubicin (DXR) and Me-ALA-based photodynamic therapy (PDT), and the non-ICD inducer cisplatin. They report a differential response to PDT in wt and BRAF V600E cell lines, as well as a differential transcriptomic profile with upregulation of IFNAR1, IFNAR2, and CXCL10 genes associated with the BRAF V600E mutation. They also provide some preliminary evidence suggesting activation of the IFN1 pathway through cGAS-STING signaling by PDT.

The topic of the work, namely the modulation of the immunologic behavior of melanoma cells as related with therapeutic options and with the mutational status of major drivers, is timely and important. The manuscript is clearly written. However, the experimental design is often not convincing, and certain important controls are missing. Overall, the results appear quite preliminary and do not completely support the interpretation provided by the authors. Accordingly, the manuscript should not be accepted, unless extensively revised.

Major points

1- Most of the reported experiments have been performed with non-isogenic cellular models. The single BRAFwt cell line (SK-MEL-2) is not isogenic with any of the two BRAF-mutated cell lines (SK-MEL-28 and A375), and these later are not isogenic with each other. Therefore, any conclusion as to the role of the BRAF status based on these cellular models is speculative and should be taken with caution.

2- Efforts should be made to present an accurate phenotypic description of the cells, at least for aspects relevant to their response to the different treatments used. It will be important to provide data on the relative expression of BRAF (by Western blot), the basal activity of the BRAF signaling pathway in the 3 cell lines (by Western blot analysis of ERK1/2 phosphorylation), and on pigmentation of the cells (ideally by measuring their melanin contents, otherwise by providing a picture of cell pellets). This is particularly important, because SK-MEL-28 and A375 cells are amelanotic, but SK-MEL-2 cells have been reported to produce melanin. Since melanin absorbs visible light of the wavelength employed for the PDT experiments, the resistance of SK-MEL-2 cells to PDT might be explained, at least partially, by a shielding effect of melanin diminishing the effective irradiation of the Me-ALA-derived photosensitizer.

3- BRAFwt SK-MEL-2 cells used as control are in fact mutated in NRAS, as they carry the activating NRAS mutation Q61R. Since NRAS is upstream of BRAF, it is unclear why these cells have been selected as controls, and why the differences in their behavior as compared to A375 or SK-MEL-28 cells are interpreted in terms of the BRAF mutation. Indeed, the BRAF signaling pathway should be constitutively activated in the 3 cell lines.

4- The major concern expressed above is further emphasized by the lack of functional data providing experimental evidence in support of the involvement of BRAF. For instance, it will be interesting to analyze the effects of inhibiting signaling downstream of BRAF-V600E with any of the available ERK1/2 inhibitors on the response of BRAF-mutated cells to DXR or PDT, and/or on their transcriptomic profiles.

5- As reported in Figure 2, the behavior of the two BRAF-V600E cell lines is quite different. Concerning DXR treatment, SK-MEL-28 behave essentially as BRAFwt SK-MEL-2 cells and markedly different from A375 cells. However, when treated with cisplatin, A375 show basically the same behavior as BRAFwt cells whereas SK-MEL-28 show a different response. Therefore, the differential response of the BRAF-V600E mutant cells to these drugs can hardly be interpreted in terms of the BRAF status. This should be discussed.

6- Throughout the manuscript, the authors use PDT as inducer of ICD. But another ICD inducer, namely doxorubicin (DXR), yields significantly different results. Accordingly, it is unclear why the authors interpret their data in terms of ICD induction. This point should be better explained.

7- The data in Figure 3 are not convincing as presented, because in Figs 3A and 3B the results of the FACS profiles do not match the corresponding bar graphs (see, for instance in Fig 3B, the large difference in the GFP signal corresponding to control and DXR-treated GFP-positive cells and compare with the minor/null difference in the bar graphs).

8- To assess the activity of the IFN1 pathway, the authors generated stable transfectants of SK-MEL-2 cells expressing the GFP protein under the control of the MX2 promoter. With this biological tool, the authors found some evidence that PDT upregulated the IFN1 pathway. However, it is unclear why SK-MEL-2 cells were chosen for this study given that this is the cell line less sensitive to the PDT treatment. Moreover, if the aim of the work is to analyze the effects of the BRAF V600E mutation, the relevant experiment would have been to compare transfectants of the MX2 promoter-GFP construct in BRAFwt versus BRAF-mutated cells.

9- The experiment described in Fig 4 to analyze involvement of the cGAS-STING pathway in PDT-induced IFN1 pathway activity is quite preliminary, as it involves exclusively one STING inhibitor which is used without providing data on the efficacy of STING inhibition and on lack of off-target effects. It will be helpful to provide stronger evidence, for instance with knockdown (shRNAs or siRNA) or knockout of selected STING pathway genes to reproduce the effect of the inhibitor.

10- The Abstract and the Conclusions sections should be extensively reworded to avoid overinterpretation of the data. Most importantly, the first two sentences in the Abstract state that the BRAF V600E mutation is associated with aggressive behavior and poor prognosis. But this is not supported by Kaplan Meier analysis of the SKCM dataset of TCGA, which shows similar survival of carriers of BRAFwt or BRAF-mutated melanomas. The manuscript should be carefully edited to avoid such overstatements.

Comments on the Quality of English Language

No major problems with the quality if Ebglish.